# Impact of Simulated Gastrointestinal Digestion on the Biological Activity of an Alcalase Hydrolysate of Orange Seed (*Siavaraze, Citrus sinensis*) by-Products

**DOI:** 10.3390/foods9091217

**Published:** 2020-09-02

**Authors:** Seyadeh Narges Mazloomi, Leticia Mora, M-Concepción Aristoy, Alireza Sadeghi Mahoonak, Mohammad Ghorbani, Gholamreza Houshmand, Fidel Toldrá

**Affiliations:** 1Instituto de Agroquímica y Tecnología de Alimentos (CSIC), Avenue Agustín Escardino 7, Paterna, 46980 Valencia, Spain; samira.mazloomi@yahoo.com (S.N.M.); mcaristoy@iata.csic.es (M-C.A.); ftoldra@iata.csic.es (F.T.); 2Department of Food Science & Technology, Gorgan University of Agricultural Sciences & Natural Resources, Gorgan 4918943464, Iran; moghorbani@yahoo.com; 3The Health of Plant and Livestock Products Research Center, Department of Pharmacology, Mazandaran University of Medical Sciences, Sari 4815733971, Iran; dr.houshmand_pharmaco@yahoo.com

**Keywords:** orange seed, bioactive peptides, antioxidant, ACE-inhibitory, antidiabetic activity, gastrointestinal digestion

## Abstract

In this study, orange seed proteins were hydrolyzed by Alcalase enzyme at different enzyme concentrations 1–3% (*v*/*w*) and hydrolysis times (2–5 h), to obtain bioactive peptides showing antioxidant, Angiotensin-converting enzyme (ACE) -inhibitory, and hypoglycemic activities. The highest biological activities (*p* < 0.05) were achieved by using a hydrolysis time of 5 h and an enzyme concentration of 2%. Orange seed protein hydrolysate (OSPH) was prepared under these conditions, and peptides were isolated and purified by using size-exclusion chromatography and high-performance liquid chromatography, respectively. The fractions that showed the highest biological activities were analyzed by mass spectrometry in tandem, and a total of 63 peptide sequences were found. Moreover, the effect of simulated gastrointestinal digestion on the bioactivity of the fractions was studied, and the novel peptide sequences generated were also identified. Overall, despite there being some differences in the profile of peptide sequences obtained, the main results showed non-significant differences in the analyzed bioactivities after simulated gastrointestinal digestion.

## 1. Introduction

Throughout the years, studies have proved that health and nutrition are extremely interrelated. Not only is food the supplier of the necessary nutrients for the correct functionality of the metabolism, but certain compounds such as hydrolyzed proteins and peptides can also stimulate and modulate specific and desirable physiological reactions in the body [1]. Bioactive peptides sequence present in the parent’s proteins needs to be further hydrolyzed to have a function; in fact, processes such as proteolytic microorganisms fermentation, plant or microorganism-extracted enzymes proteolysis, and gastrointestinal digestion can release them and activate their function [2,3]. As the use of microbial proteases is simple, non-expensive, and completely safe, it is the preferred methodology for hydrolysis processes compared with the other commercial proteases from plant and animal origin [4]. The resulting hydrolyzed proteins and peptides have been described to exert an important biological role, including antioxidant [5], antimicrobial [6], anticancer [5], antidiabetic [7], and antihypertensive activities [8]. They also influence the cardiovascular, immune, nervous, and gastrointestinal systems [8,9].

Generally, the hydrolysis of protein reduces its molecular weight, increases the ionic groups’ number, and enhances the access to hydrophilic regions [10]. To show their activities, peptides must reach their target sites mainly through the bloodstream [11]. Based on the results of experiments on the effect of gastrointestinal digestion conditions in vitro condition on the antioxidant and antimicrobial activity of hydrolyzed flaxseed proteins, it was found that gastrointestinal digestion had very little effect on the antimicrobial activity of peptides; however, this effect has been greater on antioxidant activity [12]. In a study conducted by Orsini et al. (2011), Amaranth hydrolysates were digested under simulated gastrointestinal digestion. After evaluating the antioxidant activity, they saw that the digested samples showed a good percentage of ABTS radical inhibition activity and inhibition of reactive oxygen species [13]. The results of the assay showed that simulated gastrointestinal digestion increased the Angiotensin-converting enzyme (ACE) -inhibitory activity of bean and lentil hydrolysates [14].

Hydrolyzed proteins prepared from less economically valued sources, especially food industry by-products, can be considered as a rich reservoir of bioactive peptides to be applied as nutritional supplements or functional enhancers for the production of functional foods. In this sense, *Siavaraze Citrus sinensis* of citrus species and *Rutaceae* family are commonly used in the juice production industry. Iran produces around 2,700,000 tons of citrus fruits and ranks seventh in the fruit production. Citrus species are used fresh or processed to juice; consequently, large amounts of seeds are discharged at processing plants. The wastes of the juice industry, including peels, seeds, and pulps, represent about 50% of the raw processed fruit. This potentially valuable resource, otherwise processed, can aggravate disposal problems [15]. Orange seed flour, having been defatted, is considered a by-product containing about 17.9–26.5% of protein that is possible to be implemented as a valuable and economical source for proteins extraction and hydrolysis [16]. The objective of the current study was to investigate the optimum circumstances for enzyme-driven hydrolysis of orange seed protein concentrate, using Alcalase enzyme for the production of hydrolyzed proteins showing the highest antioxidant, antihypertensive, and hypoglycemic activities. Having been separated by size-exclusion chromatography (SEC) and HPLC-RP, Alcalase hydrolysate was then investigated, and those fractions showing the highest bioactivity were analyzed, using mass spectrometry in tandem (MS/MS), in order to identify the active peptides.

## 2. Materials and Methods

### 2.1. Materials

Seeds meal of orange fruit (*Siavaraze Citrus sinensis*) with 3.6 ± 0.32% protein, 54.2 ± 12% lipid, 10.13 ± 0.65% moisture, and 2.5 ± 0.23 ash was purchased from Kosar Cultivation and Technology factory, Gorgan, Iran. Alcalase enzyme (protease from *Bacillus lichenformis*, 2.4 U/g) and all other chemicals were obtained from Merck and were of analytical grade.

### 2.2. Productions of Orange Seed Proteins Concentrate

Orange seed proteins concentrate was produced based on the method which was described by Horax et al. [17]. Distilled water was added to the orange seed flour, with the ratio of 10:1, at room temperature; they were then mixed, and pH was adjusted to 10, using 1 mol/L NaOH. The sample was subsequently stirred for 1 h, at ambient temperature, and centrifuged at 12,000 rpm, for 15 min (Avanti J-26S XP, Beckman, Brea, CA, USA). Then the pH of supernatant was adjusted to 3 by 1 mol/L HCl, and it was kept at room temperature for 30 min. The suspension was then centrifuged at 12,000 rpm, at the same temperature for 15 min. The produced pellets were washed with 20 mL distilled water and then freeze-dried (SCANVAC, Labogene ApS, Alleryd, Denmark).

### 2.3. Optimization of Enzymatic Hydrolysis

The enzymatic hydrolysis was conducted on protein solution with concentration of 0.05% *w/v*, using Alcalase enzyme at different concentrations, namely 1, 1.5, and 3% (enzyme to substrate ratio); a temperature of 55 °C; and a hydrolysis time of 2, 3.5, and 5 h, in a shaker incubator (Radleys, Essex, UK). The samples included the following: Sample 1, 0.01 E/S-2 h; Sample 2, 0.02 E/S-2 h; Sample 3, 0.03 E/S-2 h; Sample 4, 0.01 E/S-3.5 h; Sample 5, 0.02 E/S-3.5 h; Sample 6, 0.03 E/S-3.5 h; Sample 7, 0.01 E/S-5 h; Sample 8, 0.02 E/S-5 h; and Sample 9, 0.03 E/S-5 h. After deactivating the enzyme at 85 °C for 15 min, we centrifuged the sample (Suprema 25, TOMY, Tokyo, Japan) at 4 °C, for 20 min, at 12,000 rpm. The supernatant was separated and later lyophilized to be stored at −20 °C for further analysis. The measurement of the antioxidant DPPH scavenging activity and ferric-reducing power activities; the ACE-inhibitory activity; and the hypoglycemic α-amylase- and α-glucosidase-inhibitory activities were carried out and recorded for each treatment. Those conditions showing the highest bioactivity were selected and used for the production of orange seed protein hydrolysate [18].

### 2.4. Determination of Hydrolysis Degree

The hydrolysis degree was calculated according to the method adopted by Kaewka et al. (2009) [19]. A total of 10 mL protein hydrolysate and 10 mL 10% trichloroacetic acid (TCA) were mixed and centrifuged at 4 °C, for 10 min, at 10,000 rpm. Nitrogen content in supernatant and total nitrogen were measured by Kjeldahl method, and the degree of hydrolysis was calculated by using the following formula:Hydrolysis degree (%) = (Total N in supernatant/Total N in whole sample) × 100.(1)

### 2.5. Assay of DPPH Radical Scavenging Activity Measurement

In total, 100 μL of protein hydrolysate was blended with 500 μL of ethanol (96%) and 125 μL of 2,2-diphenyl-1-picrylhydrazyl (DPPH) solution (0.02% in ethanol, (*v*/*v*)); samples were then incubated for 60 min, at ambient temperature and in a dark place. Absorbance of samples was observed at 517 nm [20]. Distilled water and butylated hydroxytoluene (1 μg/μL) acted as negative and positive controls, respectively. DPPH radical scavenging activity was calculated through the following equation:Antioxidant activity (%) = (Abs of negative control − Abs of sample)/(Abs of negative control) × 100.(2)

### 2.6. Assay of Ferric-Reducing Antioxidant Power Measurement

A total of 250 μL of protein hydrolysate was mixed with the same volume of 200 mM sodium phosphate buffer, at a pH 6.6, and the same volume of 10 mg/mL of potassium ferricyanide. Next, the mixture was incubated during 20 min, at 50 °C, in the dark. Then, 250 μL of 100 mg/mL of trichloroacetic acid was added to the solution. After centrifuge at 1650 rpm for 10 min, we removed 500 μL of supernatant and added 500 μL of bi-distilled water and 100 μL of ferric chloride (1 mg/mL). Then, the sample was incubated for 10 min, at room temperature, and the absorbance was measured at 700 nm. Bi-distilled water and butylated hydroxytoluene (1 μg/μL) were used as negative and positive controls, respectively. There is a positive correlation between sample absorbance and ferric-reducing power [21].

### 2.7. Assay of the ACE-Inhibitory Activity Measurement

The assay was developed according to Sentandreu and Toldrá (2006) with some modifications. Briefly, 50 μL of protein hydrolysate was blended with 50 μL of 150 mM Tris-base buffer (pH 8.3), which contained 3 mU/mL of ACE, and then 200 μL of 150 mM Tris-base buffer (pH 8.3), which contained 1.125 M of NaCl, and then 10 mM of Abz-Gly-Phe(NO2)-Pro was added. Finally, the final product was incubated at 37 °C, for 60 min. The action of releasing o-aminobenzoylglycine as a result of ACE generated a fluorescence that was read at 355 and 405 nm as excitation and emission wavelengths, respectively. The fluorescence generated in the reaction was measured each 15 min along the incubation time. ACE-inhibitory activity was calculated as follows:ACE-inhibitory activity (%) = ((Abs Sample T60 − Abs Sample T0)/(Abs Control T60 − Abs Control T0)) × 100.(3)

Based on the results in pretests, the dilution of 1:100 (sample: distilled water) used for this assay [22].

### 2.8. Assay of α-Amylase-Inhibitory Activity

The α-amylase-inhibitory activity was assessed by using a commercial kit (SPINREACT, ref Amylase-LQ, Girona, Spain). The formation rate of 2-chloro-4-nitrophenol, which was observed photometrically, is positively correlated with the catalytic concentration of α-amylase present in the sample. Then 80 μL of protein hydrolysate solution was added to 40 μL of α-amylase enzyme (from porcine pancreas, A3176, 1 μU), and they were mixed accordingly. Then the reaction mixture was incubated at 37 °C, for 10 min. After that, 160 μL of the 2-chloro-4-nitrophenyl-α-D-maltotrioside (CNPG3) containing ((MES pH = 6, 100 mM), (CNPG3, 2.25 mM), (sodium clorhidre, 350 mM), (calcium acetate, 6 mM), (potassium thiocyanate, 900 mM), (sodium azide 0.95 gr/L)) was added, and then the reaction mixture was stored at 37 °C, for 5 min. Finally, the optical density was recorded at 405 nm. Phosphate buffer 20 mM (pH 6.8) and Acarbose (2 mg/mL phosphate buffer) were considered as blank and positive controls, respectively (SPINREACT, Girona, Spain). The α-amylase-inhibitory activity was determined as follows:α-amylase-inhibitory activity (%) = ((Abs control − Abs sample)/Abs control) × 100.(4)

### 2.9. Assay of α-Glucosidase-Inhibitory Activity

The α-glucosidase-inhibition activity was measured using a kinetic method with a commercial kit (Sigma-Aldrich, α-glucosidase Activity Assay Kit, St. Louis, MO, USA). In this assay, α-glucosidase activity was recorded through a reaction in which α-glucosidase hydrolysis of ρ-nitrophenyl-α-D-glucopyranoside results in the formation of a colorimetric substance, which subsequently can be read at 405 nm. One unit of α-glucosidase was explained to be the amount of enzyme that facilitates the hydrolysis of 1 μmol of substrate per minute at pH = 7. For the analysis, 20 μL of protein hydrolysate solution was mixed with 50 μL of α-glucosidase enzyme (from *Saccharomyces cerevisiae*, 1 mg/mL). The reaction medium was incubated at 37 °C, for 5 min, and then 200 μL of the Master Reaction Mix containing 200 μL of assay buffer (pH 7) and 8 μL of α-NPG substrate were added to the medium and incubated at 37 °C, for 60 min. Acarbose (2 mg/mL) was used as positive control. The α-glucosidase-inhibitory activity was obtained from the following equation:α-glucosidase-inhibitory activity (%) = ((Abs control − Abs sample)/Abs control) × 100.(5)

### 2.10. SEC Separation of Hydrolyzed Protein

To estimate the distribution of the peptide components in the orange seed proteins hydrolysate, a SEC separation was used based on Lassoued et al. (2015) and Jemil et al. (2016) [23,24]. For this purpose, 5 g of protein hydrolysate was stirred with 35 mL of HCl (0.01 mol/L), using a magnetic stirrer. Then three volumes of ethanol was added and mixed. The solution was then stored at 4 °C, overnight, for deproteinization. The final mixture was centrifuged at 12,000 rpm, for 20 min, at 4 °C. The ethanol was removed from the sample, using a rotary evaporator. Ultimately, the deproteinized solutions were lyophilized (SCANVAC, Labogene ApS, Alleryd, Denmark). After that, 1 g of the deproteinized powder was added to 10 mL of 0.01 N HCl solution and then filtered through 0.45 μm. Then 5 mL of the filtered solution was loaded on a Sephadex G25 column (Amersham Biosciences, Uppsala, Sweden). The separation process of peptide components was performed by providing a flow rate of 15 mL/h of filtered and degassed 0.01 N HCl, and peptide fractions were automatically accumulated in 5 mL volumes. The absorbance intensity was recorded at 254 and 280 nm, using an Ultraviolet–Visible spectrophotometer (Agilent Cary 60, Agilent Technologies, Palo Alto, CA, USA). Finally, antioxidant activity, ACE-inhibitory activity, and α-amylase- and α-glucosidase-inhibitory activities were performed separately for all fractions. Peptide fractions that showed the highest activity were pooled, lyophilized, and kept at −20 °C, to be used in further purifications.

### 2.11. Isolation of Most Active Fractions Using RP-HPLC

The peptide components separated by SEC showing the highest antioxidant activity, α-amylase, α-glucosidase, and ACE-inhibitory activity, were excessively purified, using reverse-phase HPLC. The pooled peptide fractions were mixed and filtered through a 0.45 μm filter, and 30 μL was analyzed in an HPLC system (Agilent 1100, Agilent Technologies, Palo Alto, CA, USA), which was equipped with a Symmetry C18 column (4.6 × 250 mm, 5 μm) (Waters Co., Milford, MA, USA). Two solvents, namely solvent A with TFA in bi-distilled water (0.1%, *v*/*v*) and solvent B with TFA (0.085%, *v*/*v*) in acetonitrile (ACN:bi-distilled water, (60:40, *v*/*v*)), were used as mobile phases. Both mobile phases A and B were purified with a 0.45 μm filter and degassed prior to application. Peptides were first eluted with 100% solvent A for 2 min, followed by a linear gradient from 0 to 50% of solvent B, during 50 min, at a flow rate of 1 mL/min. The obtained fractions (1 mL) were measured at 214 nm, lyophilized, and assayed for their bioactivity [23,25].

### 2.12. Simulated in Vitro Gastrointestinal Digestion

The in vitro gastrointestinal digestion assay was done according to Minekus et al. (2014), with some modifications. This method was designed to imitate the physiological conditions of the human digestive system in two phases, namely the gastric phase and intestinal phase. In the gastric phase, 2 mL of 0.01 mol/L HCl was added to 500 mg of RP-HPLC fractions (fractions showing the highest bioactivity values). Porcine pepsin (2000 U/ mL in 0.01 mol/mol HCl) and CaCl_2_ were added, to have a final concentration of 0.075 mM. Samples were incubated during 3 h, to simulate digestion at 37 °C, while they were constantly stirred, and the enzyme became inactivated by increasing the pH to 7, using 1 mol/mol NaOH. For imitating the intestinal period, different digestive enzymes, including trypsin (from hog pancreas, 100 U/mL), chymotrypsin (from bovine pancreas, 25 U/mL), porcine pancreatic α-amylase (200 U/mL), porcine pancreatic lipase (2000 U/mL), and porcine bile extract (10 mM), were added. A final concentration of 0.3 mM CaCl_2_ was also used. After 2 h, at 37 °C, the digestion process was over by heating at 95 °C for 2 min. The sample was deproteinized by adding 3 volumes of ethanol and keeping the sample at 4 °C for 20 h. After that, it was centrifuged at 12,000× *g*, at 4 °C, for 10 min. Eventually, the supernatant was dehydrated in a rotatory evaporator and then lyophilized [26].

### 2.13. Free Amino Acid Analysis

The amino acid analysis was done according to a method proposed by Flores et al. (1997), with some modifications. Then 1 mL of distilled water was added to 0.1 g of deproteinized protein hydrolysate sample, to dissolve it, and then 100 μL of this solution was blended with 850 μL of distilled water and 50 μL of internal standard (including 10 mM *N*-Leu). After that, 200 μL of the mixture was evaporated and dried, and then 15 μL of drying material (including methanol:sodium acetate 1 mol/mol: TEM, 2:2:1) was included and dried again after vortexing. Then 15 μL of derivatization material (methanol:distilled water:TEA:PITC, 7:1:1:1) was added and shaken vigorously. It stayed fixed for 20 min and then evaporated and dried again. In the next step, it was dissolved in 1 mL of 5 mM Di-sodium phosphate and centrifuged at 12,000× *g*, at 4 °C, for 10 min. Finally, the supernatant was separated and kept in a special vial for HPLC analysis. The content of amino acid within the supernatant was clarified by an Agilent 1200 HPLC system (Agilent Technologies, Palo Alto, CA, USA), which was equipped with a photodiode array detector (254 nm). The used column was a Symmetry C18 column (3.9 × 300 mm, 5 μm). In total 1 mM of each measured amino acid was used as standard. The solvent mixture had two eluents, solvent (A), including 70 mM sodium acetate, which was adjusted to pH 6.55 with 10% acetic acid and 2.5% acetonitrile; and solvent (B), including acetonitrile:water:methanol (45:40:15, *v*/*v*/*v*) [27].

### 2.14. Identification of Peptides, Using Mass Spectrometry in Tandem

The nano LC–MS/MS analysis was done with a nano-LC Ultra 1D Plus system (Eksigent of AB Sciex, CA, USA), which was coupled to a quadrupole-time-of-flight (Q ToF) TripleTOF^®^ 5600 that was equipped with a nanoelectrospray ionization source (AB Sciex Instruments, Framingham, MA, USA). The system parameters used were selected according to Mora et al. (2015) [28]. The peptide components of lyophilized orange seed protein hydrolysate obtained by RP-HPLC were dissolved in 20 μL of trifluoroacetic acid (TFA 0.1%) and concentrated and purified with Zip Tip C-18 tips (Millipore, Billerica, MA, USA). Then 5 μL of the eluted sample was injected into the mass spectrometer. Samples were concentrated beforehand on a C18 trap column (3 μ, 350 μm × 0.5 mm) (Eksigent of AB Sciex, Redwood City, CA, USA), using TFA 0.1% *v*/*v* as the mobile phase, at a flow rate of 4 μL/min. After 5 min of this concentration step, the trap column was automatically put in line with a nano-HPLC capillary column (3 μm, 75 μm × 12.3 cm, C18) from Nikkyo Technos Co, Ltd. (Tokyo, Japan), to elute the compounds. The mobile phase A was 1% formic acid in water, and mobile phase B was formed by 0.1% formic acid in 100% acetonitrile. The chromatographic conditions consisted of linear gradients from 5 to 35% of solvent B during 90 min, and then from 35 to 65% for 10 min of solvent B at a flow rate of 3 μL per minute and a temperature of 30 °C. The outlet of the capillary column was straightly attached to a nanoelectrospray ion source. The operating conditions of the Ion trapping mass spectrometry were as follows: positive polarity, capillary column with temperature of 200 °C, voltage 4.5 V, and nitrogen was used as a collision gas. The first scan was a mass scan of a mass ratio of 400 to 2500. Automatic spectrometry and generation of peak list were performed by using Mascot Distiller software. Search in databases was carried out by Mascot Daemon software in combination with Mascot interface software 2.2. In terms of searching parameters, Mascot searches were done by selecting none specific enzyme, none modifications, and mass tolerance measurement 100 ppm in the MS state and 0.6 Dalton for MS/MS ions. Identification of the origin of protein for peptides was done by using the UniProt database and NCBInr database. The BIOPEP-UWM database was used to search for similarities between the sequences identified in this study and previously published bioactive peptide sequences [15,29,30]. Only identification results showing a confidence higher of 90% were reported. The BIOPEP-UWM™ database of bioactive peptides has newly become a popular tool in the research on bioactive peptides, especially on these derived from foods and being constituents of diets that inhibit the development of chronic diseases [30].

### 2.15. Statistical Analysis

Analysis of data collected from experiments on the protein hydrolysate in optimized conditions was carried out through a completely random design by SPSS software (version 19.0, SPSS Inc., Chicago, IL, USA). Each experiment was repeated in triplicate. The Duncan’s test was also utilized at 5% significance level for comparing the mean values. Excel software (Microsoft Excel Worksheet (.xlsx), Redmond, WA, USA, 2013) was used to draw charts.

## 3. Results and Discussion

### 3.1. Antioxidant Activity

The capacity of orange seed protein hydrolysate (OSPH) as an antioxidant was evaluated based on DPPH radical scavenging activity and ferric-reducing power, and the results are presented in Figure 1: Sample 1, 0.01 E/S-2 h; Sample 2, 0.02 E/S-2 h; Sample 3, 0.03 E/S-2 h; Sample 4, 0.01 E/S-3.5 h; Sample 5, 0.02 E/S-3.5 h; Sample 6, 0.03 E/S-3.5 h; Sample 7, 0.01 E/S-5 h; Sample 8: 0.02 E/S-5 h; and Sample 9, 0.03 E/S-5 h.

The enzyme-to-substrate ratio and hydrolysis time showed significant effects (*p* < 0.05) on the DPPH scavenging activity and ferric-reducing power of OSPH. As the results show, longer times of hydrolysis (5 h in Samples 7, 8, and 9) result in the best antioxidant activities, whereas no clear trends were observed at increasing E/S ratios. A remarkable increase (*p* < 0.05) in the antioxidant activity was detected, using an enzyme concentration of 2% and hydrolysis time of 5 h with 86.4 ± 0.07% and 1.467 ± 0.1 in the DPPH radical scavenging activity and ferric-reducing power, respectively. With regard to the antioxidant, results in this study revealed that the production of antioxidant peptides increases by increasing the enzyme concentration, which is in accordance with Guérard et al. (2002) [31]. Jamdar et al. (2010) and Je et al. (2009) reported that increased hydrolysis time and enzyme concentration had an increasing effect on the antioxidant activity in peanut and tuna liver protein hydrolysates, respectively [31,32,33]. Based on amino acids analysis, hydrophobic amino acids (aromatic or branched) in the peptide sequences and the presence of one or more residues of His, Pro, Cys, Tyr, Trp, Phe, or Met amino acids increase antioxidant activity [34]. Trp plays the most important role in DPPH scavenging activity, and this is probably due to its hydrogenating role [35].

### 3.2. ACE-Inhibitory Activity

As it is shown in Figure 2, an increase in hydrolysis time and enzyme concentration affected significantly the ACE-inhibitory activity of samples. Similar to antioxidant activity, longer times of hydrolysis (5 h in Samples 7, 8, and 9) results in the best antioxidant activities, whereas no clear trends were observed at increasing enzyme/substrate (E/S) ratios. A significant increase in the ACE-inhibitory activity (*p* < 0.05) was observed, using an enzyme/substrate (E/S) ratio of 2% and a hydrolysis time of 5 h (90.2 ± 0.1%). In a previous study, an increase in the amount of peptides with molecular weight smaller than 3 kDa was observed with increasing hydrolysis times; consequently, the ACE-inhibitory activity also increased [36]. ACE-inhibitory activity seems to be related to the number of hydrophobic amino acids that are available in the peptide sequences probably because of the fact that the active site of ACE by hydrophobic peptides is more reachable [24,37,38,39]. Regarding the amino acids composition, amino acids His, Pro, Ser, Glu, and Tyr were important amino acids in peptides with ACE-inhibitory activity. Studies have revealed that hydrophobic amino acid residues such as Tyr, Ala, Leu, Val, Phe, or Trp can take action as competitive ACE-inhibitors, as they selectively bind the catalytic sites of ACE [37]. Particularly, it has been discussed that aromatic amino acids presence, such as phenylalanine, at any of the three spots that is closest to the *C*-terminal is the most appropriate situation [37]. Moreover, peptides that are the most potent antihypertensive ones include positively charged amino acids such as arginine and lysine at their *C*-terminal spot [39].

### 3.3. α-Amylase- and α-Glucosidase-Inhibitory Activity

The measurement of these activities was done for each treatment (Figure 3), and the conditions which showed the greatest inhibitory activity were selected. Best results were observed by using an enzyme concentration of 3% and hydrolysis time of 5 h for the α-amylase-inhibitory activity (41.7 ± 1.1%) and an enzyme concentration of 1% and hydrolysis time of 5h for the α-glucosidase-inhibitory activity (57.0 ± 0.6%). The α-glucosidase-inhibitory activity of OSPH in this study was higher than that of whey protein hydrolysates reported by Lacroix and Li-Chan., (2013) [40]. Regarding the amino acid composition, Ser, Asp, and Glu were important amino acids in peptides with α-amylase and α-glucosidase-inhibitory activities. It has been proved that peptides with aromatic residues (Phe, Trp, and Tyr) are considered as key factors in α-amylase- and α-glucosidase-inhibitory activities [41].

Meinert et al. (2015), Selamassakul et al. (2018), Soleymanzadeh et al. (2019), and Moayedi et al. (2016) have shown that protein hydrolysates of rice, camel milk, and tomato by-products contain peptides with antioxidant and ACE-inhibitory activities, respectively [25,42,43,44]. Jonker et al. (2011) reported that the hydrolysis of casein and whey proteins leads to generation of antidiabetic peptides that these peptides reduced blood glucose in people with type 2 diabetes [45]. Based on a research conducted by Yang et al. (2012), the addition of peptides obtained from the hydrolysis of soybean protein to the diet of rats with type 1 diabetes potentiates insulinotropic actions and improves hepatic insulin sensitivity in diabetic rats [46].

According to our outcomes, the highest bioactivities (antioxidant activities, ACE, α-amylase-inhibitory, and α-glucosidase-inhibitory activities) of peptides were observed in the ratio of 2% enzyme Alcalase to the substrate and a hydrolysis time of 5 h (*p* < 0.05). Therefore, this sample was selected for further analysis in the next steps.

### 3.4. Fractionation of OSPH by SEC

For investigating the effect of molecular mass distribution on different bioactive properties of OSPH, such as ACE-inhibitory activity, α-amylase- and α-glucosidase-inhibitory activity, and antioxidant activity, the hydrolysate was initially fractionated by using SEC, and the wavelengths of fractions were read at 254 and 280 nm. Results are illustrated in Figure 4.

OSPH showed five main peaks (Parts A, B, C, D, and E) in the molecular range of 200–70,000 Da. The first peak (Part A) includes the proteins or peptides having molecular mass between 13,000 and 70,000 Da. SEC profile also showed a peak with fragments that have low molecular weights around 1400–13,000 Da eluting after bacitracin oligopeptide standard (Part B). The next peaks (Parts C, D, and E) correspond to lower molecular weight fragments. As illustrated in Figure 5A,B the highest antioxidant activity (DPPH scavenging activity and ferric reducing power) was observed in fractions F(40–44), F(45–49), and F(50–54) with 68.86 ± 1.4%, 78.73 ± 0.6%, and 53.44 ± 0.5% in DPPH scavenging activity and 1.289 ± 0.01., 1.427 ± 0.03, and 1.619 ± 0.05 in ferric-reducing power, respectively. The highest ACE-inhibitory activity of OSPH (72.89 ± 1.3%) was reported in the fraction F(45–49) (Figure 5B). The fractions F(50–54) and F(55–59) also showed an ACE-inhibitory activity of 41.16 ± 0.8% and 42.65 ± 2.2%, respectively. Generally, peptides with low molecular weight display higher ACE-inhibitory activity [24,26]. However, the results of this research showed that the fractions F(45–49) expressed higher ACE-inhibitory activity in comparison with the fractions F(50–59), indicating that peptides with lower molecular mass do not necessarily show higher activities [25,27]. According to Figure 5C, the highest α-amylase-inhibitory activity of OSPH was seen in fractions F(40–44) and F(45–49), with an activity of 52.5 ± 0.006% and 47.7 ± 0.07%, respectively. The highest α-glucosidase-inhibitory activity was reported in fraction F(75–79). Based on these results, fraction F(40–44) and F(45–49) were selected for further purification by RP-HPLC.

Generally, an Alcalase enzyme generates peptides with different range of molecular weights due to the site of cleavage imparted by the Alcalase enzyme [46]. As a Ser protease, Alcalase, contains Asn, His, and Ser amino acids in its cleavage site. The carboxyl group in Asn binds the nitrogen in the imidazole ring of His. Other nitrogen groups in His form some bonds with the proton in the hydroxyl group of Ser, to have a hydrogen bond and create a negative charge on the oxygen atom in Ser residue. The presence of an oxygen atom with a slight-negative charge makes it possible for the Alcalase enzyme to break down proteins and produce various peptides by nucleophilic attack on the amide bonds. The non-specific action of the Alcalase enzyme during the attack on amide bonds increases its efficiency in development of hydrolysis and the production of peptides with different chain lengths [47]. Several researchers have fractionated hydrolyzed proteins and peptides by using SEC and then studied the impacts of molecular size on biological activities of the fractions [25,48,49]. It has been reported that peptide size can affect the antioxidant activity and peptides with the molecular weight of less than 3000 Da reveal the highest antioxidant power [50]. Hence, the molecular weight below 3000 Da was suggested as the most efficient peptide size in antioxidant activity [24,25,47,51]. In addition to the size of peptides, the hydrophilic–hydrophobic balance of the peptides is also an influential element in the ACE-inhibitory activity [36]. The last fractions with the lowest molecular weight in SEC separation usually include free amino acids showing lower bioactivities than other peptides [24,47]. Many experiments have shown that peptides with antidiabetic activity have hydrophobic amino acids such as Val, Trp, Leu, Ile, Pro, Phe, and Cys in their sequences [42]. In addition, protein hydrolysates containing hydrophobic amino acids may be easily absorbed [51]. It has been reported that non-saccharide compounds impart their inhibitory activities through binding the enzyme active site via hydrophobic interactions [39].

### 3.5. Purification of Peptide Fractions by RP-HPLC

RP-HPLC separation of the selected purified fractions F(40–44) and F(45–49) obtained from size-exclusion chromatography resulted in an elevated number of eluted peaks, as shown in Figure 6. The mentioned fractions were freeze-dried and assayed for their biological activities, including antioxidant, ACE-inhibitory activity, α-amylase-inhibitory activity, and α-glucosidase-inhibitory activity. In the DPPH scavenging activity assay, the fraction F22 expressed the highest antioxidant activity of 66.57% (Figure 6A). In the ferric-reducing power test, the fractions F20 and F22 showed the greatest activities, namely 0.560 and 0.544, respectively. The fraction F20 showed a maximum in ACE-inhibitory activity of 84.28% (see Figure 6B). Ultimately, the antidiabetic-potential tests revealed that F2 peptides fraction with 68.78% had the α-amylase-inhibitory activity, and F28 peptides fraction with 64.70% peptides fraction had maximum α-glucosidase-inhibitory activity (see Figure 6C). According to the results of this section, Fractions 19 to 20, obtained from RP-HPLC, were chosen for sequencing and peptide characterization.

Generally, reversed-phase (RP) preparative HPLC is considered to be one of the best methods to isolate and purify peptides from hydrolyzed proteins, based on the hydrophobicity of the peptides [51]. It is clear that there was a relationship between possible increase of either DPPH or ferric-reducing power with relative increase in the hydrophobicity. Peptides with ACE-inhibitory activity need to connect to the active site of ACE or an inhibitor site placed on the ACE, adjusting the conformation of the protein [52]. Lassoued et al. (2016) declared that high hydrophilic peptides have less accessibility to the ACE active site [53].

### 3.6. Free Amino Acid Composition

The free amino acid composition of orange seed protein hydrolysate is shown in Table 1. As displayed in Table 1, the total amount of free amino acids of orange seed protein hydrolysate was 2.6657 mg/mL. The major free amino acids presented in OSPH were Arg, followed by Tyr, Trp, His, Phe, and Met. It is clear that there is not a high amount of free amino acids in OSPH (2.6657 mg/g of sample hydrolysate). Alcalase enzyme is an endonuclease that facilitates free amino acids formation in the hydrolyzed proteins [38,49]. Enzymatic hydrolysis in longer hydrolysis times and higher enzyme concentrations could simply result in a faster augmentation in the content of free amino acids. Generally, the estimation of free amino acids is frequently used as an evaluation of the intensity of hydrolysis [38].

### 3.7. Influence of Simulated In Vitro Gastrointestinal Digestion on the Bioactivity of Peptides

Chromatograms of peptide fractions F(40–44) and F(45–49) and their potential bioactivity (antioxidant, ACE, α-amylase-inhibitory, and α-glucosidase-inhibitory activities) were compared before and after digestion. The results obtained after imitative gastrointestinal digestion on bioactivity of selected fractions are shown in Figure 7. Each fraction was analyzed by RP-HPLC after simulated gastrointestinal digestion, and then their respective bioactivities were investigated.

According to the results, after digestion, in F(19–20), DPPH scavenging activity decreased from 62.56 to 36.16% and ferric-reducing power also decreased from 0.560 to 0.172 (*p* < 0.05). A significant decrease in α-amylase-inhibitory activity (from 60.30 to 39.55%) and β-glucosidase-inhibitory activity (from 56.76 to 41.22%) was also observed. This decrease might be due to digestion of some active peptides (Figure 7A,C). However, the relative percentage of Part B in Figure 7 for F(19–20) increased from 84.28 to 90.52% in the ACE-inhibitory activity after digestion. This demonstrated that, during digestion, new peptides showing important ACE-inhibitory activity were generated as well. Generally, this result shows that fraction F20 was more resistant than other fractions against gastrointestinal digestion and, therefore, was selected for the next step.

Elimination of some peaks, changes in the time of withdrawal of some other peaks from the column after gastrointestinal digestion, and reduction of the following levels of a number of peaks in digested fractions indicated the digestive effect of proteases enzymes on changes in the structure of peptide chains. In addition, treatment in acidic conditions with pH = 2 and then increasing pH to 7 causes changes in peptide chains. The results implicated that the structure of F(19–20) of the peptide had less affected by gastrointestinal digestion than other fractions [54].

Meshginfar et al. (2018) reported that the net charge of peptide sequence influences its solubility and likely its digestion [54]. In another study, amaranth antioxidant peptides (antioxidant activity was investigated by the ABTS^+^ scavenging and the ORAC assays) and tomato antioxidant peptides (antioxidant activity was measured by DPPH scavenging activity) showed minimal changes in their bioactivities after in vitro gastrointestinal digestion [13,54].

### 3.8. Identification of Peptide in Most Active HPLC Fractions by MS/MS

Fractions 19 and 20, obtained from RP-HPLC, were chosen for characterization by nano LC–MS/MS analysis prior to and post simulated gastrointestinal digestion, and the corresponding results are shown in Table 2 and Table 3. Generally, 63 peptide sequences were identified in all samples before simulated digestion (Table 2). The size of the peptides in this study, before and after simulated gastrointestinal digestion, had 6–24 and 7–18 amino acids in length, respectively, whereas the analysis of poultry protein hydrolysate after in vitro gastrointestinal digestion on the ACE-inhibitory activity resulted in ACE-inhibitory peptides from five to nine amino acids in length [55].

With regard to the amino acid composition, His, Pro, Ser, Asp, and Glu were the main amino acids existing in the peptide sequences with antioxidant activity investigated in OSPH. On the other hand, the amino acids His, Pro, Ser, Glu, and Tyr and the amino acids Pro, Ser, Asp, and Glu were the most important amino acids present in the peptides with ACE-inhibitory and α-amylase- and α-glucosidase-inhibitory activities, respectively; this results are similar to those of the study conducted by Maqsoudlou et al. (2018), Meshginfar et al. (2018), who reported that these amino acids and other hydrophobic amino acids were the principal amino acids in the peptide sequences of pollen and tomato protein hydrolysate, respectively [49,54]. Mora et al. (2015) claimed that the ACE-inhibitory peptides contain Lys, Pro, or aromatic residues preferably in the three spots nearest to the *C*-terminal section [28]. Regarding the amino acid mixture in the bioactive peptide sequences in OSPH, after gastrointestinal digestion process, His, Pro, Ser, Asp, Glu, and Tyr were the leading amino acids existing in the peptides.

Bioactivity of hydrolyzed proteins and peptides depends on the hydrolysis situations, protein of origin, degree of hydrolysis, molecular weight, and amino acid sequences [47]. Identified peptides from OSPH in this research included peptides with molecular weights, between 900 and 3000 and between 900 and 2000 Da, before and after gastrointestinal digestion, respectively. Moayedi et al. (2016), Maqsoudlou et al. (2018), and Lassoued et al. (2015) recorded the same sizes for the ACE-inhibitory and antioxidant peptides [25,49]. Salema et al. (2018) announced that the peptides identified from protein hydrolysates from *Octopus vulgaris* by nano-liquid chromatography and mass spectrometry in tandem after RP-HPLC separation, with the highest antidiabetic and anti-hyperlipidemic effects, were composed of molecular weights around 400–2500 Da, and the majority of them included many hydrophobic amino acid residues [56]. Liu et al. (2014) investigated the antioxidant activity of peptides from porcine plasma protein hydrolysate prepared by Alcalase enzyme and showed that fractions with molecular weight less than 3 kDa exhibited the greatest DPPH scavenging activity and ferric-reducing power [57]. Elimination of some peaks, changes in exit time of some other peaks from the column, and reduction of the following levels of some peaks indicated the digestion effect of proteases in the structural change of peptide chains. Treatment in acidic conditions also reduces some peaks in high-molecular-weight peptide chains in the chromatogram chart, and peptide sequences appear at about 1 kDa and less [54]. The biological activity of peptides has been defined to be highly dependent on the amino acids composing the sequence, the hydrophobicity and molecular weight of the molecule, and the length of amino acids. Some bioactive peptides are resistant to the action of proteases, so they can remain intact after ingestion and further gastrointestinal digestion and bloodstream transport to the target sites [54,55].

Thus, the impacts of molecular weight and amino acid mixture on the antioxidant, α-amylase inhibitory- and α-glucosidase-inhibitory activities, and ACE-inhibitory activity showed that both of these two elements play a vital role in the bioactivity of peptides generated during protein hydrolysis. The fractions 19 to 20 obtained from RP-HPLC were chosen to be identified, and a total of 63 peptide sequences were recognized in all samples. According to results obtained after imitative gastrointestinal digestion, DPPH scavenging activity, ferric-reducing power, α-amylase-inhibitory activity, and α-glucosidase-inhibitory activity decreased after digestion, and ACE-inhibitory activity increased. These results showed significant differences in any of the biological activities studied after digestion (*p* < 0.05). Thus, orange seed proteins could be employed as a new protein source for the manufacturing of peptides with antioxidant ability, ACE-inhibitory activity, and α-amylase-inhibitory and α-glucosidase-inhibitory activities with relative resistance to gastrointestinal digestion. These peptides could be applied in different food formulations and protein-based food supplements, to improve human health.

## 4. Conclusions

This study was conducted to evaluate antioxidant, ACE-inhibitory activities and antidiabetic capacity, as well as to study the peptides stability of hydrolyzed orange seed proteins by Alcalase enzyme. Our results indicated that the maximum biological activities, especially antioxidant and ACE-inhibitory activities, of the hydrolyzed orange seed proteins were obtained at hydrolysis time of 5 h and the enzyme to substrate ratio of 0.02. Based on the SEC and RP-HPLC tests, we observed that both content and molecular weight of the amino acids play a very important role in the biological activity of the obtained peptides. The peptides fractions (19–20) derived from RP-HPLC were selected to sequence and identify peptides. The molecular weight of the peptides before and after simulated gastrointestinal digestion was 900–3000 and 900–2000 Da, respectively. Elimination of some peaks, changes in the time of withdrawal of some other peaks from the column after gastrointestinal digestion, and reduction of the following levels of a number of peaks in digested fractions indicated the digestive effect of proteases enzymes on changes in the structure of peptide chains. It seems that orange seeds can be used to produce beneficial bioactive peptides that are resistant to digestive enzymes. Therefore, Alcalase hydrolyzed orange seed proteins can be suggested as health beneficial product to reduce blood pressure and diabetes management. Future studies especially randomized clinical trials are warranted to obtain a precise conclusion about health beneficial effects of orange seed proteins hydrolysate.

## Figures and Tables

**Figure 1 foods-09-01217-f001:**
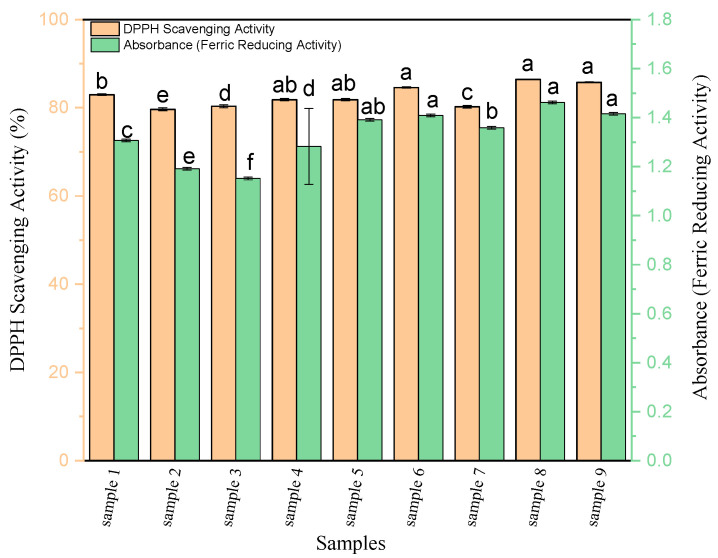
DPPH scavenging activity (%) and ferric-reducing power activity of tomato protein hydrolysate (OSPH) under different conditions. Data are mean ± SD of three replications and the values for each activity with different letter are significantly different (*p* < 0.05).

**Figure 2 foods-09-01217-f002:**
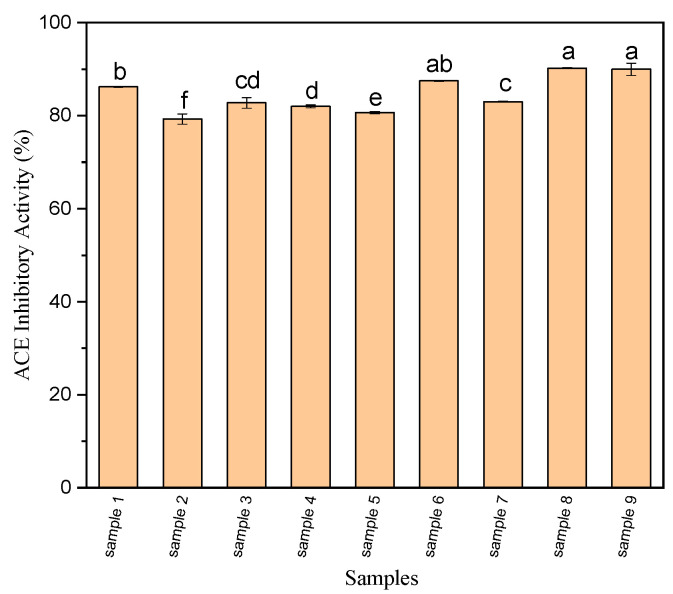
Angiotensin-converting enzyme (ACE) inhibitory activity (%) of OSPH produced under different conditions. Data are mean ± SD of three replications and the values with different letter are significantly different (*p* < 0.05).

**Figure 3 foods-09-01217-f003:**
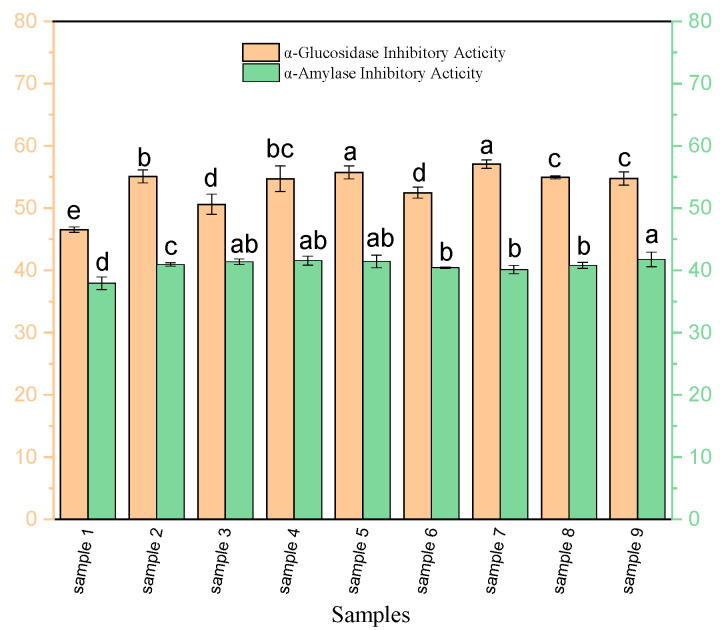
Overall α-amylase- and α-glucosidase-inhibitory activity (%) of OSPH under different conditions. Data are mean ± SD of three replications and the values for each activity with different letter are significantly different (*p* < 0.05).

**Figure 4 foods-09-01217-f004:**
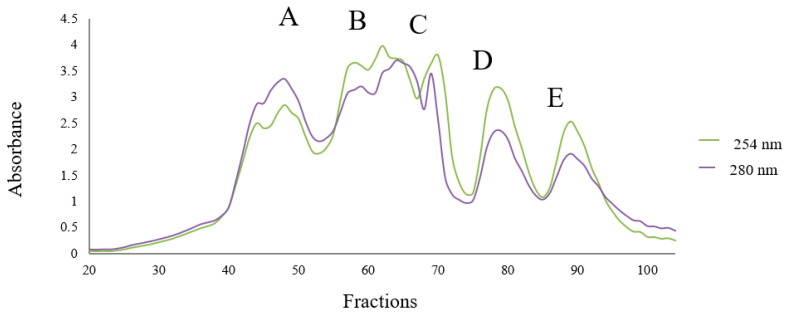
Size-exclusion chromatography (SEC) profile of OSPH (0.1 g/mL), using a Sephadex G-25 column.

**Figure 5 foods-09-01217-f005:**
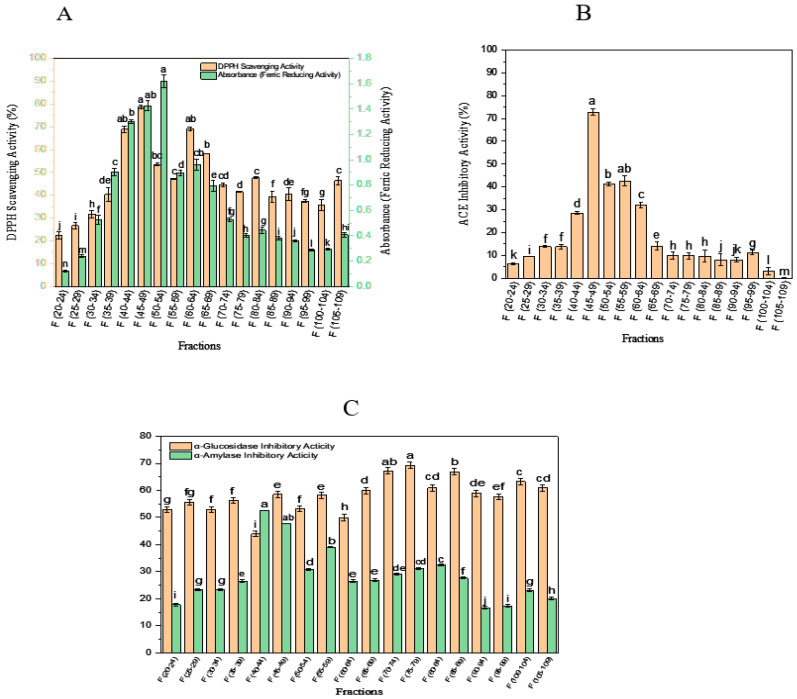
DPPH scavenging activity and ferric-reducing power (**A**), ACE-inhibitory activity (**B**), and α-amylase-inhibitory activity and α-glucosidase-inhibitory activity (**C**) in different fractions obtained from size-exclusion chromatography of OSPH. F(40–44) and F(45–49) were selected for further purification. Data are mean ± SD of three replications and the values with different letter for each activity are significantly different (*p* < 0.05).

**Figure 6 foods-09-01217-f006:**
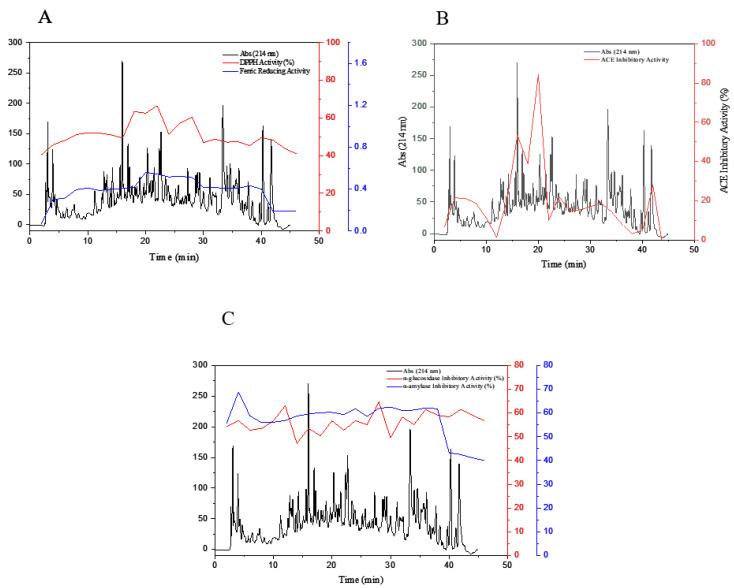
Reversed-phase high-performance liquid chromatography (RP-HPLC) separation of Fractions (40–44) and (45–49) obtained from SEC. The fractions were automatically collected and assayed for their DPPH scavenging activity and ferric-reducing power (**A**), ACE-inhibitory activity (**B**), and α-amylase-inhibitory activity, and α-glucosidase-inhibitory activity (**C**). Fractions 19 to 20, obtained from RP-HPLC, were selected for further sequencing and identification.

**Figure 7 foods-09-01217-f007:**
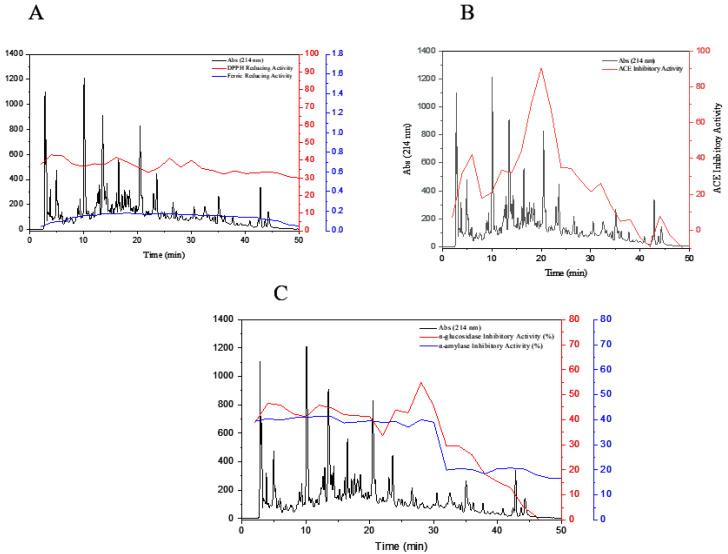
Reversed-phase chromatographic separation of F(40–44) and F(45–49) from SEC and after in vitro gastrointestinal digestion. Fractions were automatically collected and assayed for their DPPH scavenging activity and ferric-reducing power (**A**), ACE-inhibitory activity (**B**), and α-amylase-inhibitory activity and α-glucosidase-inhibitory activity (**C**). Fractions 19 and 20 obtained from RP-HPLC were selected for sequencing and identification.

**Table 1 foods-09-01217-t001:** Free amino acid composition of orange seed protein hydrolysate.

Amino Acid Name	Concentration (mg/g of OSPH) ± SD
**Asp**	0.1276
**Glu**	0.1434
**Ser**	0.1087
**Asn**	0.13 ± 0.0001
**Gly**	0.0742
**Gln**	0.1278 ± 0.0001
**β Ala**	0.0885 ± 0.0001
**His**	0.1616 ± 0.0001
**Thr**	0.1172
**Ala**	0.0871 ± 0.0003
**Arg**	0.1682
**Pro**	0.1103
**Tyr**	0.1665
**Val**	0.1158
**Met**	0.1439
**Ile**	0.1245
**Leu**	0.1305
**Phe**	0.153
**Trp**	0.1637
**Orn**	0.0812
**Lys**	0.1417
**Total**	2.6657 ± 0.001

**Table 2 foods-09-01217-t002:** Differential peptide sequences present in F(19–20) identified from OSPH (before gastrointestinal (GI) digestion), using NCBInr and UniProt database.

Sequence	Protein	Bioactivity (Sequence Previously Described as Responsible for the Bioactivity)	MW
ATREQRQQQQRFQTQ	Uncharacterized protein OS = *Citrus clementina* tr|V4S993|V4S993_9ROSI	Inhibition of prolyl endopeptidase (IHPFAQTQ)	1933.3
EQEQEFQGSGD	Uncharacterized protein OS = *Citrus clementina* tr|V4S993|V4S993_9ROSI	ACE-inhibitory (RKSGDPLGR)	1253.36
HNINDPSGA KHNINDPSGA	Uncharacterized protein OS = *Citrus clementina* tr|V4S993|V4S993_9ROSI	Antimicrobial activities (GYVSGAVIEIPDEILDSAR)	924.06 1052.51
HNINDPSGAD KHNINDPSGAD	Uncharacterized protein OS = *Citrus clementina* tr|V4S993|V4S993_9ROSI	Celiac toxic (GWFGGADWHA)	1039.16 1166.53
HNINDPSGADA	Uncharacterized protein OS = *Citrus clementina* tr|V4S993|V4S993_9ROSI	Kinases inhibitor, (KKALRRQEAADAL) Antioxidative (LLPHHADADY) Antithrombotic, Antimicrobial (DNIADAVACAKRVVRDPQGIR)	1110.25
PSGADAYNPR	Uncharacterized protein OS = *Citrus clementina* tr|V4S993|V4S993_9ROSI	Antioxidative (PRHVFYRWFLSNPRI)	1052.25
QESQQRSSESQSRSQDQHQKVR	Uncharacterized protein OS = *Citrus clementina* tr|V4S993|V4S993_9ROSI	Antimicrobial (TKCFQWQRNMRKVRGPPVSCIKR)	1167.35
QQQQRFQTQ TREQRQQQQRFQTQ	Uncharacterized protein OS = *Citrus clementina* tr|V4S993|V4S993_9ROSI	ACE inhibitor (QTQSLVYP)	1047.21 1903.93
SSESQSRSQDQHQKVR FQSSKSQDQHQKVR	Uncharacterized protein OS = *Citrus clementina* tr|V4S993|V4S993_9ROSI	Antimicrobial (AGRGKQGGKVRAKAKTRSSRA) ACE inhibitor (KVREGTTY)	2644.05 1701.84
LRHNIDKPSHAD	Uncharacterized protein OS = *Citrus clementina* tr|V4S993|V4S993_9ROSI	Antimicrobial (LLPHHADADY)	1191.41
QDSQQQQSFQSS	Uncharacterized protein OS = *Citrus clementina* tr|V4S993|V4S993_9ROSI	Neuropeptide (FSEFMRQYLVLSMQSSQ)	1887.18
SFQSSKSQDQHQKVR	Uncharacterized protein OS = *Citrus clementina* tr|V4S993|V4S993_9ROSI	ACE inhibitor (KVREGTTY)	1862.21
SQGGRSQGSQGSDDRRAGN SQGSQGSDDRRAGN	Citrin OS = *Citrus sinensis* Q39627_CITSI Uncharacterized protein OS = *Citrus clementina* tr|V4U2L8|V4U2L8_9ROSI	VLKMAGNSFQEN (Celiac Toxic coeliac toxic peptide) YLAGNQ, EVMAGNYLPG, EVMAGNLYPG (ACE inhibitor)	1918.83 1433.59
SQSQGGRSQGSQGSDDRRAGNL GSQGSDDGRGGNL SQGGRSQGSQGSDDGRGGNL SQSQGGRSQGSQGSDDGRGGNL	Citrin OS = *Citrus sinensis* Q39627_CITSI	Antimicrobial (GLFDAIGNLLGGLGLG) Antiamnestic (PEP inhibitor) (RYDWWPYGNLFGGHTFISP)	2247.01 1218.51 1918.83 2133.91
VFPGCAETFQDSQQQQSFQSSKSQDQHQKVR	Uncharacterized protein OS = *Penicillium digitatum* tr|K9FZ18|K9FZ18_PEND1	Antimicrobial (AGRGKQGGKVRAKAKTRSSRAGLQFPV) ACE inhibitor (KVREGTTY)	3549.70
VTDITREGKQQ	Citrin OS = *Citrus sinensis* tr|Q39627|Q39627_CITSI	Regulating (QQQKQQQQPSSQVS)	1790.14
GTQDHPHDDYAE	Uncharacterized protein OS = *Citrus clementina* tr|V4S993|V4S993_9ROSI	Anti-inflammatory peptide (KGHYAERVG) ACE inhibitor (YAEERYPIL)	1920.17
GTQDHPHDDYAEAK	Uncharacterized protein OS = *Citrus clementina* tr|V4S993|V4S993_9ROSI	Kinases inhibitor (VTCDILSVEAKGVKLG)	1434.59
AGDTHLGGED	Uncharacterized protein OS = *Penicillium digitatum* tr|K9FZ18|K9FZ18_PEND1	Antithrombotic (EAGEDCDCGSPANPCCDAATCKLIPGAQCGEGLCCDQCSFIEEGTVCRIARGDDLDDYCNGRSAGCPRNPFH)	2135.41
DQGNRITPS	Uncharacterized protein OS = *Penicillium digitatum* tr|K9FZ18|K9FZ18_PEND1	Antimicrobial (GIFSSRKCKTPSKTFKGICTRDSNCDTSCRYEGYPAGDCKGIRRRCMCSKPC) Neuropeptide (NKLASVYALTPSLRVG) TPSPR (Alpha-glucosidase inhibitor)	2248.59
SEGTEPIQSK GQKDAYVGDEAQSK	Protein disulfide-isomerase OS = *Citrus limon* tr|V9HXG3|V9HXG3_CITLI	Antimicrobial (FVPYNPPRPGQSKPFPSFPGHGPFNPKIQWPYPLPNPGH)	3585.3
DYDKPVQQ	Uncharacterized protein OS = *Citrus clementina* tr|V4S993|V4S993_9ROSI	Coeliac toxic peptide (QQPFVQQQQPFVQQ)	1219.38
GADDSADNKSSNAPTRTY	Uncharacterized protein OS = *Citrus clementina* tr|V4UJF9|V4UJF9_9ROSI	Erythropoietin receptor agonist peptide (YQRRPAIAINNPYVPRTYYANPAVVRPHAQIPQRQYLPNSHPPTVVRRPNLHPSF)	1920.2
GETGGPHPGYETR	Uncharacterized protein OS = *Citrus clementina* tr|V4SQX0|V4SQX0_9ROSI	Alpha amylase inhibitory (DETRL)	2135.44
IVPRKAASSEE	Uncharacterized protein OS = *Citrus clementina* tr|V4S6W0|V4S6W0_9ROSI	Antioxidative (RELEELNVPGEIVESLSSSEESITR)	1274.56
EVHNPATGE	Succinate-semialdehyde dehydrogenase OS = *Citrus clementina* tr|V4RMN0|V4RMN0_9ROSI	ACE inhibitor (VLSPPFTGE)	1384.5
QDQGPMVK	Uncharacterized protein OS = *Citrus unshiu* tr|A0A2H5NKL3|A0A2H5NKL3_CITUN	Neuropeptide (SPTISITAPIDVLRKTWEQERARKQMVKNREFLNSLN) Membrane-active peptides (MVKSKIGSWILVLFVAMWSDVGLCKKRPKP)	1583.78
GQMNEPPGAR	ATP synthase subunit beta (Fragment) OS =*Poncirus trifoliata* tr|Q9THW1|Q9THW1_PONTR	Antimicrobial, (GICACRRRFCPNSERFSGYCRVNGARYVRCCSRR)Opioid, (GICACRRRFCPNSERFSGYCRVNGARYVRCCSRR, YGGFTGARKSARKLANQ, FGGFTGARKSA) Neuropeptide (YGGFLGARKSARKLANQ) Antioxidative (QGAR)	971.09
FYLGGNPQPQLQ	Uncharacterized protein OS = *Citrus clementina* tr|V4S993|V4S993_9ROSI	Neuropeptide (MPRVRSLFQEQEEPEPGMEEAGEMEQKQLQ) coeliac toxic peptide (LQLQPFPQPQLPYPQPQLPYPQPQLPYPQPQPF)	987.16
QQQRFQTQ	Uncharacterized protein OS = *Citrus clementina* tr|V4S993|V4S993_9ROSI	ACE inhibitor (QTQSLVYP)	1075.28
GSAKESGDKAEQGS	ABC transporter solute-binding protein OS = *Streptomyces* sp. tr|A0A3L8QSZ7|A0A3L8QSZ7_9ACTN	coeliac toxic peptide, (QQQQPSSQVSFQQPLQQYPLGQGSFRPSQQNPQA, QGSFRPSQQNPQAQ, QYPLGQGSFRPS, LGQGSFRPSQQN) ACE inhibitor, (YQGS) Antioxidative (AWEEREQGSR)	992.16
SQSRSQDQHQKVRQIRE	Uncharacterized protein OS = *Citrus clementina* tr|V4S993|V4S993_9ROSI	Neuropeptide (SENFTPWAYIILNGEAPIIREVHYSPRL)	1870.1
HNIDKPSHAD	Uncharacterized protein OS = *Citrus clementina* tr|V4U2L8|V4U2L8_9ROSI	Antioxidative (LLPHHADADY)	1495.76
QSFQSSKSQDQHQKVR	Uncharacterized protein OS = *Citrus clementina* tr|V4U2L8|V4U2L8_9ROSI	ACE inhibitor (KVREGTTY, KVREGT)	1357.59
QDSQQQQSFQS	Uncharacterized protein OS = *Citrus clementina* tr|V4U2L8|V4U2L8_9ROSI	Antimicrobial (GLLSRLRDFLSDRGRRLGEKIERIGQKIKDLSEFFQS)	1186.47
QPAGKRGEGPPKRAGQVR	REVERSED Uncharacterized protein OS = *Curtobacterium* sp. RRRRRtr|A0A1E5MMN3|A0A1E5MMN3_9MICO	Celiac toxic (VQGQGHQPPQQPAQL, VQGQGIIQPQQPAQL)	953.1
QKVESEAGVT	Uncharacterized protein OS = *Citrus clementina* tr|V4U2L8|V4U2L8_9ROSI	Antioxidative (DSGVT)	902.15
VIGTVLAFALIASAESLF	Sulfate transporter OS = *Streptomyces* sp. tr|A0A3L8QBZ9|A0A3L8QBZ9_9ACTN	Antimicrobial (SLFSLIKAGAKFLGKNLLKQGAQYAACKVSKECG, SLFSLIKAGAKFLGKNLLKQGACYAACKASKQC) Neuropeptide (MPRVRSLFQEQEEPEPGMEEAGEMEQKQLQ)	1056.3
TAAEEVLQKAEAAP	CHAD domain-containing protein OS = *Methylobacterium mesophilicum* tr|M7YUM6|M7YUM6_9RHIZ	Antimicrobial (WNPFKELERAGQRVRDAVISAAPAVATVGQAAAIARG)	1361.71
SAEKGNLYQN	Uncharacterized protein OS = *Citrus clementina* tr|V4S993|V4S993_9ROSI	Regulating (EPFYQNVPD)	1063.26
NALEPR	Uncharacterized protein OS = *Citrus clementina* tr|V4U2L8|V4U2L8_9ROSI	Peptide stimulating insulin release Antioxidative (GFGSFLGKALKA ALKIGANALGGSPQQ)	1350.55
QNPANPVTLAQAIEGEPK	DNA-directed DNA polymerase OS = *Methylobacterium mesophilicum* tr|M7Y0I8|M7Y0I8_9RHIZ	ACE Inhibitor (EPKAIP, HSGIQSEPKAIP)Antioxidative (APIRMWYMYRKLTDMEPKPVA)	2110.53
SSDTIDNVKAK	Polyubiquitin OS = *Avena fatua* UBIQP_AVEFA	Antimicrobial (GCASRCKAKCAGRRCJGWASASFRGRCYCKCFRC)	1133.32
QSDVTTTS	Transcription factor MYB21 OS = *Arabidopsis thaliana* MYB21_ARATH	Antimicrobial (QLKTADLPAGRDETTSFVLV) Antioxidative (PIAAEVYEHTEGSTTSY)	1918.29
DESTGTIGKR	Fructose-bisphosphate aldolase, cytoplasmic isozyme 1 OS = *Pisum sativum* ALF1_PEA	Neuropeptide (KRQHPGKR)	1310.46
SAPKKEKSQGFLQ	Type II inositol polyphosphate 5-phosphatase 14 OS = *Arabidopsis thaliana* IP5PE_ARATH	ACE Inhibitor, DPP IV inhibitor (TPVVVPPFLQP, FLQP)	1219.35
FASTRRRQCPQ	Lignin-forming anionic peroxidase OS = *Nicotiana sylvestris* PERX_NICSY	Antimicrobial (CPQLQPQNPSQQQPQEQG)	1889.41
GSGAKPGTKPTKCT	Chaperone protein dnaJ A6, chloroplastic OS = *Arabidopsis thaliana* DNJA6_ARATH	Antimicrobial (RSVCRQIKICRRRGGCYYKCTNRPY)	1047.27
TNADKATTVS	Soluble starch synthase 3, chloroplastic/amyloplastic OS = *Solanum tuberosum* SSY3_SOLTU	Antimicrobial (GILDTLKQFAKGVGKDLVKGAAQGVLSTVSCKLAKTC, GIFSSRKCKTVSKTFRGICTRNANC)	1822.44
WTAEHSVNAALGQFE	Pyruvate, phosphate dikinase regulatory protein, chloroplastic OS = *Zea mays* PDRP1_MAIZE	Celiac Toxic (QQLPQFEEIRNL, GSVQPQQQLPQFEIR)	1427.76
GSEEPNVEEDS	FHA domain-containing protein DDL OS = *Arabidopsis thaliana* DDL_ARATH	Antimicrobial (YPGPQAKEDSEGPSQGPASREK)	1123.33

Notes: Peptides are grouped and aligned according to their sequence. MW: molecular weight.

**Table 3 foods-09-01217-t003:** Differential peptide sequences present in F(19–20) identified from OSPH (after GI digestion), using NCBInr and UniProt database.

Sequence	Protein	Bioactivity	MW
SQGSQGSDDGRGGNL	Citrin OS = *Citrus sinensis* Q39627_CITSI	PEP inhibitor (Antiamnestic), (RYDWWPYGNLFGGHTFISP) Antibacterial, (LLPIVGNLLKSLL, GLLDMVTGLLGNLG, GLFDAIGNLLGGLGLG)Antioxidative, (QLGNLGV)ACE inhibitor (EVMAGNLYPG)	1219.38
GSQGSDDGRGGNL	Citrin OS = *Citrus sinensis* Q39627_CITSI	PEP inhibitor (Antiamnestic), (RYDWWPYGNLFGGHTFISP)Antibacterial, (LLPIVGNLLKSLL, GLLDMVTGLLGNLG, GLFDAIGNLLGGLGLG) Antioxidative, (QLGNLGV)ACE inhibitor (EVMAGNLYPG)	1434.62
GSQGSDDGRGGNL	Citrin OS = *Citrus sinensis* Q39627_CITSI	PEP inhibitor (Antiamnestic), (RYDWWPYGNLFGGHTFISP)Antibacterial, (LLPIVGNLLKSLL, GLLDMVTGLLGNLG, GLFDAIGNLLGGLGLG)Antioxidative, (QLGNLGV)ACE inhibitor (EVMAGNLYPG)	1219.38
SQGSQGSDDRRAGN	Uncharacterized protein OS = *Citrus clementina* tr|V4U2L8|V4U2L8_9ROSI	Celiac Toxic, (VLKMAGNSFQEN) ACE inhibitor (YLAGNQ, EVMAGNYLPG, EVMAGNLYPG)	1434.59
IDLPQPQ	Serine/threonine-protein kinase TOR OS = *Arabidopsis* TOR_ARATH	ACE inhibitor, (QPQPLIYP) Potential coeliac toxic peptide, (PQNPSQQQPQEQVP) Celiac Toxic, (SQPQAFP) Antibacterial (PYPQPQPF)	1219.38
RLAIEEAISITTTLVAQY	Protein RETICULATA-RELATED 5, chloroplastic OS = *Arabidopsis thaliana* RER5_ARATH	Antibacterial, (NLLKQGAQYAACKVSKECG) ACE inhibitor (VLAQYK)	1434.59
SSCPVINVD	Uncharacterized GPI-anchored protein At1g61900 OS = *Arabidopsis thaliana* UGPI6_ARATH	Antibacterial, (CSCRTSSCRFGERL) Natriuretic (RSSCFGGRIDRIGAC)	810.02
RHSWMMN	Phosphoenolpyruvate carboxylase kinase 2 OS = *Arabidopsis thaliana* PPCK2_ARATH	Antibacterial, (VPMPKGRSSRGRRHS) Antioxidative (RHS)	1992.57

Peptides have been grouped and aligned according to their sequence.

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
