# Peer review of "Impact of Simulated Gastrointestinal Digestion on the Biological Activity of an Alcalase Hydrolysate of Orange Seed (Siavaraze, Citrus sinensis) by-Products"

_foods, 2020, doi:10.3390/foods9091217_

Round 1
Reviewer 1 Report
Regarding the manuscript “Impact of simulated gastrointestinal digestion on the biological activity of an alcalase hydrolysate of orange (sivaraze citrus sinensis) by-products” I wish to make the following general remarks: The paper presents some novelty since there is no previous work on the biopeptides derived from orange seed protein concentrate. The manuscript is very interesting and worth publishing. However, in my opinion, corrections need to be made before final acceptance of the manuscript for publication.
GENERAL REMARK: the text should be grammatically checked (e.g line 32); mandatory use SI units; captions under the figures and tables: add necessary information that will allow to analyze data without reading the whole text, add units used were they are omitted.
Section INTRODUCTION
the health-promoting importance of plant proteins as well as importance of gastrointestinal digestion should be emphasized. This will be a good reference to the subject of the manuscript.
Section MATERIAL AND METHODS
Line 167 – add some details concerning „molecular range” estimation
Line 186 – you used currently available (from 2018) BIOPEP-UWM database instead of BIOPEP database (the previous one); cite the proper literature reference.
Section RESULTS AND DISCUSSION
Line 306 – cite references concerning biopeptides structure-activity relationship, chemometrics and chemoinformatics analysis etc.
Line308 – ACE instead od ACEI.
Line 329-330 – the sentence is not clear.
Line 338-339 – the sentence is not clear. Add additional arguments and expand the explanation.
Line 343-392 – figure 5 is not discussed; Figure 6 concerns RP-HPLC separation.
Line 369 - add additional arguments and expand the explanation.
Line 407-408 – figure B concerns ACE inhibitory activity and C – antidiabetic activity.
Line 411 – superscript?
Line 418 – selected collected – unnecessary words.
Line 430 - the sentence is not clear.
Line 436 – join columns 2 and 3.
Line – 440 – what fractions; what bioactivity.
Line 444-445 - the sentence is not clear.
Line 451 - add additional arguments and expand the explanation.
Line 471 - the sentence is not clear.
Line 472 – give the types of bioactivity
Line 509 – remove column „Database” and add relevant information under the table; add column concerning the type of biological activity; the following properties: amphipathicity, hydroplicity and charge can be deeper discussed.
Line 527 - considering the final results, the sentence is not clear.
Author Response
GENERAL REMARK: the text should be grammatically checked (e.g line 32); mandatory use SI units; captions under the figures and tables: add necessary information that will allow to analyze data without reading the whole text, add units used were they are omitted.
Thanks for valuable reviewer comment. The text has been checked and corrected.
Section INTRODUCTION:
*the health-promoting importance of plant proteins as well as importance of gastrointestinal digestion should be emphasized. This will be a good reference to the subject of the manuscript.
Thanks for valuable reviewer comment. It was modified according to referee opinion. More explanation is given in the introduction.
Section MATERIAL AND METHODS
*Line 167 – add some details concerning „molecular range” estimation.
Thanks for valuable reviewer comment, Standard proteins used for column calibration include bovine serum albumin (70 kDa), cytochrome C (13 kDa), Aprotinin (6511.44 Da), Bacitracin (1.42 kDa), Carnosine (0.22 kDa) and Tyrosine (0.2 kDa).
*Line 186 – you used currently available (from 2018) BIOPEP-UWM database instead of BIOPEP database (the previous one); cite the proper literature reference.
Thanks for valuable reviewer comment. It was modified according to referee opinion:
BIOPEP-UWM database. Minkiewicz P., Iwaniak A., Darewicz M., 2019. BIOPEP-UWM Database of Bioactive Peptides: Current Opportunities.
International Journal of Molecular Sciences, 20, 5978, doi: 10.3390/ijms20235978.
Section RESULTS AND DISCUSSION
*Line 306 – cite references concerning biopeptides structure-activity relationship, chemometrics and chemoinformatics analysis etc.
Thanks for the comment; we tried to show this item in the result and discussion section. We also showed the relationship between bioactivity of peptides, their structure and amino acid sequence in the section.
*Line308 – ACE instead of ACEI
Thanks for valuable reviewer comment. It was modified according to referee opinion.
*Line 329-330 – the sentence is not clear.
Thanks for valuable reviewer comment, It was modified according to referee opinion.
* Line 338-339 – the sentence is not clear. Add additional arguments and expand the explanation.
Thanks for valuable reviewer comment. It was modified according to referee opinion.
*Line 343-392 – figure 5 is not discussed; Figure 6 concerns RP-HPLC separation.
Thanks for valuable reviewer comment. in fact figures 4 and 5 are discussed in 3-4 section. Figure 6 has been modified.
*Line 369 - add additional arguments and expand the explanation.
The text has been modified and expanded.
* Line 407-408 – figure B concerns ACE inhibitory activity and C – antidiabetic activity.
Thanks for valuable reviewer comment, It was modified according to referee opinion.
*Line 411 – superscript?
Thanks for valuable reviewer comment, It was modified.
*Line 418 – selected collected – unnecessary words.
Thanks for valuable reviewer comment, It was modified according to referee opinion.
* Line 430 - the sentence is not clear.
Thanks for valuable reviewer comment. It was modified according to referee opinion.
*Line 436 – join columns 2 and 3.
Thanks for valuable reviewer comment. It was modified according to referee opinion.
*Line – 440 – what fractions; what bioactivity.
Thanks for valuable reviewer comment. It was modified according to referee opinion.
*Line 444-445 - the sentence is not clear.
Thanks for valuable reviewer comment. It was modified according to referee opinion.
* Line 451 - add additional arguments and expand the explanation.
Thanks for valuable reviewer comment. It was modified according to referee opinion.
*Line 471 - the sentence is not clear.
Thanks for valuable reviewer comment. It was modified according to referee opinion.
*Line 472 – give the types of bioactivity.
Thanks for valuable reviewer comment. It was modified according to referee opinion.
*Line 509 – remove column „Database” and add relevant information under the table; add column concerning the type of biological activity; the following properties: amphipathicity, hydroplicity and charge can be deeper discussed.
The changes have been made according to referee opinion. The amphipathicity, hydroplicity and charge have been omitted because the length of paper in too long and just important data presented and discussed.
*Line 527 - considering the final results, the sentence is not clear.
Thanks for valuable reviewer comment. The sentence has been modified.

Reviewer 2 Report
The manuscript by Mazloomi et al, descibes the bioactivity of different orange seed hydrolysates, their fractions and different digests. Even though a great amount of work has been done, the manuscript and results are unfortunately poorly written, presented and interpreted – more work should be put in proofreading and taking the reader by the hand. Especially I would prefer a more thorough investigation of the fractions produced, besides just their bioactivity.
Major comments
· Introduction: please provide more information about the enzyme selected for hydrolysis, what kind of enzyme is this? Why this enzyme compared to other options? What are the orange seeds currently used for and do you know how much of this by-product is generated?
· Line 259: How well described are the orange seed proteins? To state that you use the UniProt database is too unspecific. Which modifications was included? what about mass range? Did you include both Swiss-Prot and TrEMBLE databases?
· Section 3 is results and discussion, not only results.
· The results of the measured bioactivity of the samples is poorly described. Minor effects are only found. Only the samples with highest activity is described. No considerations are provide besides that. What about tendencies in the Enzyme/substrate ration and hydrolysis time?
· Table 2 and 3: Which proteins does these peptides derive from? Also at which pH is this the expected charge? In your text you dont describe anything about the Amphipathicity and the Hydrophilicity and the charge of the identified peptides but you include it in the table?
· What do you use the BIOPEP search for as described in the MM section? I don’t see that in the result.
· From your peptide identification, which peptides are likely to exert the effect? What are your purpose with this data? To make it useful it would be nice to compare with the peptide profile of a fraction with low biological activity.
Minor comments:
· Line 34: “also can” change to “can also”
· Line 35-36: I don’t understand the sentence “The mentioned bioactive peptides are not often active in their original structure,” it sound like you mean that the bioactive peptides needs to be further hydrolyzed to have a function, is this true?
· Line 45-46: Unclear, when you hydrolyse a protein, it is not a protein anymore so refereeing back to it losing molecular weight is confusing.
· Loine 60-61: confusing.
· Line 83: Why was this very low protein concentration solution used?
· Line 171: you use proteins, but you want to separate peptides in the hydrolysate, should you use the protein hydrolysate in your SEC seperation instead of the protein?
· Line 172: Why do you deproteinize?
· Line 275: Its combined in figure 1. But I would prefer you make it into two figures for easier interpretation.
· 275-277: Would be nice to include in the M&M section as well
· Figure 1: State what the letters compare, is it between the two activities or between samples?
· Figure 1: I think you can in a smart way show what the samples are in the figures instead of writing sample XX. Writing “Sample 1” is just as long as writing “0.01 E/S- 2h”
· Line 283: I wouldn’t say its considerable effect to the DPPH scavenging activity! And I don’t really see any clear pattern for the ratio.
Author Response
Reviewer #2:
The manuscript by Mazloomi et al, descibes the bioactivity of different orange seed hydrolysates, their fractions and different digests. Even though a great amount of work has been done, the manuscript and results are unfortunately poorly written, presented and interpreted – more work should be put in proofreading and taking the reader by the hand. Especially I would prefer a more thorough investigation of the fractions produced, besides just their bioactivity.
Major comments:
*Introduction: please provide more information about the enzyme selected for hydrolysis, what kind of enzyme is this? Why this enzyme compared to other options? What are the orange seeds currently used for and do you know how much of this by-product is generated?
Thanks for valuable reviewer comments. Alcalase enzyme in this research was a bacterial protease, subtilisin A from Bacillus species (Bacillus lichenformis, 2.4 U/g). Based on our previous studies andusing error and trail the Alcalase showed the best activity in producing bioactive peptides and therefore was selected for further studies.
Current annual worldwide citrus production is estimated at over 70 million tons, with more than half of this being oranges. Iran produces two million and 700,000 tons of self-sustaining production and ranks seventh in the production of citrus fruits.
The objective of the study was to hydrolyze proteins prepared from less economically valued sources especially food industry by-products, which can be used as nutritional supplements, functional enhancers for production of functional food and as rich source of bioactive peptides with good biological activities. Iran produces around 2,700,000 tons of citrus fruits and ranks seventh in the fruit production. Citrus species are used fresh or processed to juice; consequently large amounts of seeds are discharged at processing plants. The wastes of juice industry include peels, seeds and pulps represent about 50% of the raw processed fruit. This potentially valuable resource, otherwise processed can aggravate disposal problems. The defatted seed flour as a by-product of juice industry contains about 17.9-26.5% protein and can be used as a rich and cost-effective source for the production of proteins, hydrolyzed proteins and peptides of plant origin.
All above sentences included in the paper.
*Line 259: How well described are the orange seed proteins? To state that you use the UniProt database is too unspecific. Which modifications was included? what about mass range? Did you include both Swiss-Prot and TrEMBLE databases?
Citrus sinensis proteins are only partially described in the protein databases so all Green plant proteomes were used for the identification of the peptides of interest. Two different searches were done using UniProt database (including SwissProt and TrEMBLE) and NCBInr database, by selecting none specific enzyme, mass tolerance measurement of 100 ppm in the MS state and 0.6 Da for MS/MS ions, any posttranslational modification was selected. The mass range was established during the MS/MS analysis in the instrument parameters to scan a mass ratio of 400 to 2500. Only identification results showing a confidence higher of 90% were reported.
*Section 3 is results and discussion, not only results.
Thanks for comment. It was modified according to referee opinion.
*The results of the measured bioactivity of the samples is poorly described. Minor effects are only found. Only the samples with highest activity is described. No considerations are provide besides that. What about tendencies in the Enzyme/substrate ration and hydrolysis time?
The tendencies in the E/S ratio and hydrolysis times have been described according to the biological activity.
*Table 2 and 3: Which proteins does these peptides derive from? Also at which pH is this the expected charge? In your text you dont describe anything about the Amphipathicity and the Hydrophilicity and the charge of the identified peptides but you include it in the table?
The protein of origin has been included on tables 2 and 3. The amphipathicity, hydroplicity and charge have been removed from the manuscript and results obtained after the use of BIOPEP database have been included instead.
*What do you use the BIOPEP search for as described in the MM section? I don’t see that in the result.
Tables 2 and 3 have been completed with more specific data obtained from BIOPEP database. Some fragments of identified peptides were previously described as bioactive peptides, and the sequences of the terminal di and tripeptides as well as their bioactivity have been included on Tables.
* From your peptide identification, which peptides are likely to exert the effect? What are your purpose with this data? To make it useful it would be nice to compare with the peptide profile of a fraction with low biological activity.
Peptides identification gives a large list of peptides that could (or not) be exerting biological activity. Thus, the obtained sequences were searched in bibliography and and bioactive peptides databases in order to confirm their potential as bioactives. This data has been included in Tables 2 and 3.
Minor comments:
*Line 34: “also can” change to “can also”
Thanks for comment, It was modified according to referee opinion.
*Line 35-36: I don’t understand the sentence “The mentioned bioactive peptides are not often active in their original structure,” it sound like you mean that the bioactive peptides needs to be further hydrolyzed to have a function, is this true?
Thanks for valuable reviewer comment, the sentence has been modified for a better understanding.
* Line 45-46: Unclear, when you hydrolyse a protein, it is not a protein anymore so refereeing back to it losing molecular weight is confusing.
Thanks for valuable reviewer comment, the sentence has been modified for a better understanding.
*Line 60-61: confusing.
Thanks for valuable reviewer comment, It was modified according to referee opinion.
* Line 83: Why was this very low protein concentration solution used?
Thanks for valuable reviewer comment. This concentration has been selected according to previous experiments as well as saving the initial amount of protein.
* Line 171: you use proteins, but you want to separate peptides in the hydrolysate, should you use the protein hydrolysate in your SEC seperation instead of the protein?
Thanks for the comment, it was a typic mistake,the text is related to protein hydrolysate and according to your suggestion, the text has been modified.
*Line 172: Why do you deproteinize?
Thanks for valuable reviewer comment, in fact during initial hydrolysis, the Alcalase enzyme may not be able to break some of the strong bonds of the primary protein, so some proteins may not be hydrolysed due to stronger bonds. Therefore we do deproteinization to separate unhydrolyzed proteins from hydrolyzed proteins and peptides.
*Line 275: Its combined in figure 1. But I would prefer you make it into two figures for easier interpretation.
Thanks for valuable reviewer comment. There are too many figures so we have to put in one figure from 1 to 5 by combining 1+2, 3, 4+5 in the same graphic using secondary axis.
*275-277: Would be nice to include in the M&M section as well.
Thanks for valuable reviewer comment. It was modified according to referee opinion.
*Figure 1: State what the letters compare, is it between the two activities or between samples?
Thanks for comment, it is between samples.
*Figure 1: I think you can in a smart way show what the samples are in the figures instead of writing sample XX. Writing “Sample 1” is just as long as writing “0.01 E/S- 2h”
Thanks for valuable reviewer comment, in fact, there are 9 samples, when we wrote “0.01 E/S-2h and others, we need a long place in the table, therefore we have to use name abbreviation.
* Line 283: I wouldn’t say its considerable effect to the DPPH scavenging activity! And I don’t really see any clear pattern for the ratio.
Thanks for valuable reviewer comment. The sentence was modified (considerable changed to meaningful). The main objective of optimizing hydrolysis condition is to find optimum condition to achieve the highest antioxidant activity and we did not expect any pattern, on the other hand there is not any certain pattern in this regards.

Reviewer 3 Report
Review of “Impact of Simulated Gastrointestinal Digestion on the Biological Activity of an Alcalase Hydrolyzate of Orange (Siavaraze Citrus sinensis) By-products“
Overall Recommendation: Accept after minor revision
General Comments:
This manuscript reports the biological activity (antioxidant, ACE-inhibitory and hypoglycemic acitivities) of the peptide fractions obtained upon hydrolysis of orange seed proteins by Alcalase enzyme at different enzyme concentrations and hydrolysis times. The simulated gastrointestinal digestion was not found to cause significant influence on the bioactivity of the most potent peptide fractions.
In general, the description of the methodology is detailed and supported with the appropriate references and the results obtained are well elaborated.
My specific remarks are presented below:
- Title: I would like the authors to reconsider if they should specify which orange by-products were analyzed (pulp, seed or peel).
- Figures: It is difficult to get relative information from figures due to the font size being too small.
- Reference 32 cannot be found in the reference list.
- Tehnical corrections:
- 5, lines 206 and 212: change "CaCl2" to "CaCl2"
- 5, line 222: change "N-Leu" to "N-Leu"
- 7, lines 311 and 313, p. 14, line 484: change "C-terminal" to "C-terminal"
Based on the above considerations, the current form of this paper is qualified for publication and requires only minor revision.
Author Response
Reviewer #3:
Review of “Impact of Simulated Gastrointestinal Digestion on the Biological Activity of an Alcalase Hydrolyzate of Orange (Siavaraze Citrus sinensis) By-products“
Overall Recommendation: Accept after minor revision
General Comments:
This manuscript reports the biological activity (antioxidant, ACE-inhibitory and hypoglycemic acitivities) of the peptide fractions obtained upon hydrolysis of orange seed proteins by Alcalase enzyme at different enzyme concentrations and hydrolysis times. The simulated gastrointestinal digestion was not found to cause significant influence on the bioactivity of the most potent peptide fractions.
In general, the description of the methodology is detailed and supported with the appropriate references and the results obtained are well elaborated.
My specific remarks are presented below:
* Title: I would like the authors to reconsider if they should specify which orange by-products were analyzed (pulp, seed or peel).
Thanks for valuable reviewer comment, It was modified according to referee opinion.
*Figures: It is difficult to get relative information from figures due to the font size being too small.
Thanks for valuable reviewer comment. In fact, high-quality shapes are drawn with Origin software and no matter how much you zoom in, the quality will not go down.
* Reference 32 cannot be found in the reference list.
Thanks for valuable reviewer comment, It was modified according to referee opinion.
Tehnical corrections:
- 5, lines 206 and 212: change "CaCl2" to "CaCl2"
Thanks for valuable reviewer comment, It was modified according to referee opinion.
- 5, line 222: change "N-Leu" to "N-Leu"
Thanks for valuable reviewer comment, It was modified according to referee opinion.
- 7, lines 311 and 313, p. 14, line 484: change "C-terminal" to "C-terminal"
Thanks for valuable reviewer comment, It was modified according to referee opinion.
Based on the above considerations, the current form of this paper is qualified for publication and requires only minor revision.
